# Tracking inflammation resolution signatures in lungs after SARS-CoV-2 omicron BA.1 infection of K18-hACE2 mice

Agnes Carolin[1‡], Kexin Yan[1‡], Cameron R. Bishop[1‡], Bing Tang[1], Wilson Nguyen[1], Daniel J. Rawle[1‡]*, Andreas Suhrbier[1,2‡]*

1 QIMR Berghofer Medical Research Institute, Brisbane, Queensland, Australia, 2 GVN Centre of Excellence, Australian Infectious Disease Research Centre, Brisbane, Queensland, Australia

‡ AC, KY, and CRB contributed equally as joint first authors. DJR and AS contributed equally as joint last authors
* Andreas.Suhrbier@qimrberghofer.edu.au (AS); Daniel.Rawle@qimrberghofer.edu.au (DJR)

**Data Availability Statement:** All relevant data are included within the article and its supporting information files. The raw sequencing data (FASTQ files) generated from the RNA-Seq analysis have

## Abstract

The severe acute respiratory syndrome coronavirus 2 (SARS-CoV-2) causes Coronavirus Disease 2019 (COVID-19), which can result in severe disease, often characterised by a 'cytokine storm' and the associated acute respiratory distress syndrome. However, many infections with SARS-CoV-2 are mild or asymptomatic throughout the course of infection. Although blood biomarkers of severe disease are well studied, less well understood are the inflammatory signatures in lung tissues associated with mild disease or silent infections, wherein infection and inflammation are rapidly resolved leading to sequelae-free recovery. Herein we described RNA-Seq and histological analyses of lungs over time in an omicron BA.1/K18-hACE2 mouse infection model, which displays these latter features. Although robust infection was evident at 2 days post infection (dpi), viral RNA was largely cleared by 10 dpi. Acute inflammatory signatures showed a slightly different pattern of cytokine signatures compared with severe infection models, and where much diminished 30 dpi and absent by 66 dpi. Cellular deconvolution identified significantly increased abundance scores for a number of anti-inflammatory pro-resolution cell types at 5/10 dpi. These included type II innate lymphoid cells, T regulatory cells, and interstitial macrophages. Genes whose expression trended downwards over 2–66 dpi included biomarkers of severe disease and were associated with 'cytokine storm' pathways. Genes whose expression trended upward during this period were associated with recovery of ciliated cells, AT2 to AT1 transition, reticular fibroblasts and innate lymphoid cells, indicating a return to homeostasis. Very few differentially expressed host genes were identified at 66 dpi, suggesting near complete recovery. The parallels between mild or subclinical infections in humans and those observed in this BA.1/K18-hACE2 mouse model are discussed with reference to the concept of "protective inflammation".

been deposited in the NCBI Sequence Read Archive (SRA) under BioProject ID PRJNA1081927 and are publicly accessible.

**Funding:** The authors thank the Brazil Family Foundation (and others) for their generous philanthropic donations that helped set up the PC3 (BSL3) SARS-CoV-2 research facility at QIMR Berghofer MRI, as well as ongoing research into SARS-CoV-2, COVID-19 and long-COVID. A.S. is supported by the National Health and Medical Research Council (NHMRC) of Australia (Investigator grant APP1173880). The funders had no role in study design, data collection and analysis, decision to publish, or preparation of the manuscript.

**Competing interests:** The authors have declared that no competing interests exist.

## Introduction

The severe acute respiratory syndrome coronavirus 2 (SARS-CoV-2) has caused a global pandemic of Coronavirus Disease 2019 (COVID-19) [1], with over 770 million cases and 7 million deaths [2,3]. Severe acute disease is primarily characterised by infection of the respiratory tract and pathological hyper-inflammatory responses, often referred to as a "cytokine storm" that can lead to acute respiratory distress syndrome (ARDS) and ultimately mortality [4–6]. Post-acute and chronic sequelae of COVID-19 are now well described, with post-COVID conditions affecting 10–30% of COVID-19 patients [7] and long-COVID affecting at least 10% of patients [8–10].

The inflammatory mediators that are associated with severe COVID-19, as distinct from mild disease, have been the focus of many studies, with such insights having the potential to identify therapeutic anti-disease interventions [4,6,11–18]. Elevated mRNA or protein levels of a range of cytokines have been identified in peripheral blood of severe COVID-19 patients, these include *inter alia* IL-6, IL-1, IL-2, IL-17, IL-18, TNF, IFNG and CSF2 (GM-CSF) (*ibid*). Elevated IL-10 in the peripheral blood has also been associated with severe disease, but may be useful later in the infection to suppress the hyper-cytokinemia associated with ARDS [19–21]. Elevated Th2 [11,22–24] and Th17 responses [11,25] have been associated with poor outcomes, with Th1/Th2 [26–28] and Th1/Th17 imbalances [29] also reported to be detrimental. T regulatory cells (Tregs) are viewed as providing protection against the cytokine storm [30], whereas neutrophils promote the latter and contribute to disease severity [11]. Monocytes and macrophages play key protective roles against SARS-CoV-2 infection, but depending on their differentiation and activation state, are also important contributors to over-production of inflammatory cytokines [31,32]. Higher viral loads appear to positively associate with COVID-19 severity [33,34], although this may be associated with more prolonged high viral loads in patients with severe disease [35].

An estimated 25–30% of SARS-CoV-2 infections are asymptomatic throughout the course of infection [36,37], with a constellation of factors potentially contributing [38–40]. Although fatal disease is associated with higher viral loads, asymptomatic individuals often do not have lower acute viral loads than symptomatic patients [41–44]. Humans have also been shown to tolerate a certain level of lower respiratory tract infection (and associated radiographic abnormalities), without reporting symptoms [45]. Persistent asymptomatics do, nevertheless, appear able to clear the virus more rapidly [46,47]. For instance, only minimal PCR-detectable viral RNA was present in saliva of a small cohort of such individuals by days 9–10 post first positive diagnosis (after daily saliva testing) [46]. Individuals with silent infections may have lower levels of circulating proinflammatory cytokines [47], although this is not a universal finding [48]. Silent or subclinical infections have been associated with powerful antiviral responses [49], early and robust innate immune responses [50], specific HLA genes [40], and specific gene expression patterns [51,52], such as higher plasma levels of the anti-inflammatory cytokines IL-10, IL-1RA and IL-19 [48]. Although in humans there can be long-term consequences of asymptomatic SARS-CoV-2 infections, these are significantly more prevalent after symptomatic infections [53].

Herein we provide the transcriptional profile of lung infection over time in a model of SARS-CoV-2 omicron BA.1 infection of K18-hACE2 mice, wherein the majority of mice showed no significant overt signs or symptoms of disease [54]. The acute phase was characterised by a significant lung infection and a pattern of proinflammatory cytokine signatures differing slightly from those reported previously [55]. By 10 days post infection (dpi), virus had been largely cleared and a series of cell types associated with damage control, suppression of inflammation, homeostasis and repair were identified by cellular deconvolution. Few

differentially expressed genes (DEGs) were identified in lungs by 66 dpi, arguing that the model describes near complete resolution of the infection, without long-term pulmonary sequelae. The term "protective inflammation" has been coined to describe the inflammatory processes that lead to subclinical or mild disease, resulting in sequelae-free outcomes [56]. Protective responses are likely to have distinct gene expression patterns and modulated cellular infiltrates, when compared with inflammatory disease [40,57,58]. Herein we provide insights into the transcriptional signatures in lungs that characterise resolution of inflammation and return to homeostasis, and also identify a series of parallels with human studies.

## Materials and methods

### Ethics statements and regulatory compliance

Collection of nasal swabs from consented COVID-19 patients was approved by the QIMR Berghofer Medical Research Institute Human Research Ethics Committee (P3600). Patients self-nominated to provide samples and declared they had COVID-19, with no medical records accessed. Patients took their own nasal swab samples as per rapid antigen test. Patients signed a consent form that was countersigned by the study leader. Consent forms are held on file at QIMR Berghofer MRI. Samples were deidentified and virus isolated from the nasal swab material. All participants were adults with degree-level education. Signing of consent forms was not witnessed due to infection risk.

All mouse work was conducted in accordance with the Australian code for the care and use of animals for scientific purposes (National Health and Medical Research Council, Australia). Mouse work was approved by the QIMR Berghofer MRI Animal Ethics Committee (P3600). All infectious SARS-CoV-2 work was conducted in the BioSafety Level 3 (PC3) facility at the QIMR Berghofer MRI (Department of Agriculture, Fisheries and Forestry, certification Q2326 and Office of the Gene Technology Regulator certification 3445). Breeding and use of GM mice was approved under a Notifiable Low Risk Dealing (NLRD) Identifier: NLRD_Suhrbier_Oct2020: NLRD 1.1(a). Mice were euthanized using carbon dioxide.

### The SARS-CoV-2 isolates

The BA.1 omicron isolate was obtained at QIMR Berghofer MRI from nasal swabs from a consented COVID-19 patient by culture in Vero E6 cells (ATCC C1008). The omicron BA.1 isolate, SARS-CoV-2$_{QIMR01}$ (SARS-CoV-2/human/AUS/QIMR01/2022), belongs to the BA.1.17 lineage (GenBank: ON819429 and GISAID EPI_ISL_13414183) [59,60]. BA.1 viral stocks were propagated in Vero E6 cells as described [61]. UV inactivation of the BA.1 virus was undertaken using the UVC 500 Ultraviolet Crosslinker (Hoefer) (dose 7650 J/m$^2$ UVC) as described [61]. Virus stocks were titered using cell culture infectious dose 50% (CCID$_{50}$) assays [62].

The original strain isolate, SARS-CoV-2$_{QLD02}$, (hCoV-19/Australia/QLD02/2020) (GISAID accession EPI_ISL_407896) was kindly provided by Dr. Alyssa Pyke (Queensland Health Forensic &Scientific Services, Queensland Department of Health, Brisbane, Australia). Propagation and titration was undertaken as for BA.1.

### K18-hACE2 mice, infection and monitoring

Heterozygous K18-hACE2 mice (strain B6.Cg-Tg(K18-ACE2) 2Prlmn/J, JAX Stock No: 034860) were purchased from The Jackson Laboratory, USA, and were maintained in-house as heterozygotes by backcrossing to C57BL/6J mice (Animal Resources Center, Canning Vale WA, Australia). Heterozygotes were inter-crossed to generate homozygous K18-hACE2 mice (on a C57BL/6J background [63]) as described [54]. Mice were held under standard animal

house conditions (for details see [59]) and homozygous female mice received intrapulmonary infections delivered via the intranasal route with $5\times10^4$ $CCID_{50}$ of virus in 50 μl RPMI 1640 whilst under light anesthesia. Mice were anesthetized with 4% isoflurane (Piramal Enterprises Ltd., Andhra Pradesh, India) administered using The Stinger, Rodent Anesthesia System (Advanced Anaesthesia Specialists/Darvall, Gladesville, NSW, Australia), oxygen flow rate 0.8 L/min. Mice were placed individually into the induction chamber and after loss of righting reflex (≈30 sec) the mice were kept in the chamber for 2 mins before intranasal administration of virus. For each group for each time point, a comparable mean age and age distribution of mice was used (mean age 26. 3 ± SD 1.1. weeks). Mice were weighed and monitored as described [54,64]. Mice were euthanized using $CO_2$, lungs were removed, with the left lung fixed in formalin for histology, the right lung inferior lobe placed in RNAlater for RNA-Seq and RTqPCR, and the remaining lobes used for tissue titers determination by $CCID_{50}$ assays using Vero E6 cells as described [62,64].

## $CCID_{50}$ assays

Tissue titers were determined as described [61]. Briefly, 5 fold serial dilutions of clarified tissue homogenates were applied in duplicates to Vero E6 cells in 96 well plates. After 6 days cytopathic effects were observed by inverted light microscope [62]. The virus titer was determined by the method of Spearman and Karber [65].

## SARS-CoV-2 genome copy number determination

The right inferior lung lobes were harvested and placed in RNAlater (Invitrogen). Samples were transferred to TRIzol (Life Technologies) and were homogenized twice at 6000 x g for 15 sec (Precellys 24 Homogenizer, Bertin Instruments, Montigny-le-Bretonneux, France) as described [64]. Homogenates were centrifuged at $14,000 \times g$ for 10 min and RNA was isolated as per manufacturers' instructions. cDNA was synthesized using ProtoScript II First Strand cDNA Synthesis Kit (New England Biolabs) and qPCR performed using iTaq Universal Probes Kit (Bio-Rad). Lung samples were obtained at 2 and 5 dpi from K18-hACE2 mice infected with BA.1 or an original strain isolate (SARS-CoV-2$_{QLD02}$). Primers that span the junction between Orf1a and Orf1b were used in order to measure genomic, rather than subgenomic RNA levels [66]; F 5′-GGCCAATTCTGCTGTCAAATTA-3′, R 5′-CAGTGCAAGCAGTTTGTGT AG-3′. Primers for the house keeping gene, mRPL13a, were 5′-GAGGTCGGGTGGAAGTAC CA-3′ and 5′-TGCATCTTGGCCTTTTCCTT-3′ [67]. PCR fragments of SARS-CoV-2 Orf1ab and mRPL13a were gel purified and 10-fold serial dilutions of estimated copy numbers were used as standards in qPCR to calculate copies in samples reactions. SARS-CoV-2 Orf1ab copies were normalized by mRPL13a copy number in each reaction. qPCR reactions were performed in duplicate and averaged to determine the copy number in each sample.

## Histology

Lungs were fixed in 10% formalin, embedded in paraffin, and sections stained with H&E (Sigma Aldrich). Slides were scanned using Aperio AT Turbo (Aperio, Vista, CA, USA). Areas with overt leukocyte infiltrates (with high focal densities of dark purple staining nuclei) were measured manually using the "Pen Tool" in the Aperio ImageScope software v10 (Leica Biosystems, Mt Waverley, Australia). White space analysis was undertaken using QuPath v0.2.3 [68].

## Immunohistochemistry

Immunohistochemistry was undertaken as described using the anti-SARS-CoV-2 spike monoclonal antibody, SCV2-1E8 [60], except that the monoclonal (IgG2a) was purified using Protein A affinity chromatography and applied to sections at 4 μg/ml for 1 hr.

## RNA isolation, library preparation, RNA-Seq and bioinformatics

RNA isolation, library preparation and RNA-Seq was undertaken as described [55,69]. Briefly, lung tissues were harvested into RNAlater, RNA was extracted using TRIzol (Life Technologies), and RNA concentration and quality measured using TapeStation D1kTapeScreen assay (Agilent). cDNA libraries were generated using Illumina TruSeq Stranded mRNA library prep kit. RNA-Seq was undertaken as described using Illumina Nextseq 2000 platform generating 75 bp paired end reads [55]. Mean quality scores were above Q20 for all samples. Mouse RNA-Seq reads were aligned to a combined mouse (GRCm39, version M27) and SARS-CoV-2 BA.5 reference genome [54] using STAR aligner. Viral read counts were generated using Samtools v1.16. Read counts for host genes were generated using RSEM v1.3.1. Genes with low read counts were identified and removed separately for each time point using DESeq2 v1.40.2. Differentially expressed host genes were identified using DESeq2 using a FDR cut-off of q<0.05. Pathway analysis was performed with host DEGs using Ingenuity Pathway Analysis (IPA, v84978992) (QIAGEN) using the Canonical pathways, Up-Stream Regulators (USR) and Diseases or Functions features as described [63]. Gene Set Enrichment Analyses (GSEAs) were undertaken using GSEA v4.1.0 with gene sets provided in MSigDB ($\approx$ 45,000 gene sets) and in the Blood Transcription Modules [70], and gene lists generated using DESeq2 ranked by $\log_2$ fold-change. Relative abundances of cell types were estimated in R v4.1.0 from RSEM 'expected counts' using SpatialDecon v1.4.3 [71] and cell-type expression matrices obtained from the "Mouse Cell Atlas Lung Cell expression matrix" and the "NanoString Immune Cell Family expression matrix".

## RTqPCR validation of RNA-Seq data

The RTqPCR was conducted as described [69]. Briefly, cDNA was synthesized from total RNA with ProtoScript II First Strand cDNA Synthesis Kit (New England Biolabs) and qPCR performed using iTaq Universal SYBR Green Supermix (Bio-Rad) as per manufacturer's instructions with primers (Integrated DNA Technologies) for mouse *Oas3* (Forward 5'-TGGCAATC CCATCAAGCCAT-3' and Reverse 5'- CTGAGGGCTGGTGTCACTTT-3'), *Irf7* (Forward 5'-AC CGTGTTTACGAGGAACCC-3' and Reverse 5'-GTTCTTACTGCTGGGGCCAT-3'), *Ccl8* (Forward 5'- GGGTGCTGAAAAGCTACGAGAG-3' and Reverse 5'- GGATCTCCATGTACTCACT GACC-3'). qPCR reactions were performed in duplicate and averaged. Gene expression was normalized with the house-keeping gene, *mRpl13a* [67]. The 2-ΔΔCt method was used to calculate the $\log_2$ fold-change [72].

## Statistics

The t-test (with Welch's correction) was used if the difference in variances was <4 fold, skewness was > - 2 and kurtosis was <2. The t test significance and variance were determined using Microsoft Excel. Skewness and kurtosis were determined using IBM SPSS Statistics for Windows v19.0 (IBM Corp., Armonk, NY, USA). Otherwise, the non-parametric Kolmogorov-Smirnov exact test was performed using GraphPad Prism 10.

## Results

### SARS-CoV-2 omicron BA.1 provided robust lung infection in K18-hACE2 mice that was cleared by 10 dpi

K18-hACE2 mice were given an intrapulmonary inoculum via the intranasal route of SARS-CoV-2 BA.1 ($5x10^4$ $CCID_{50}$) or the same inoculum of UV-inactivated BA.1 virus. Lungs were harvested at 2, 5, 10, 30 and 66 days post infection (dpi) and tissue titers (Fig 1A), RNA-Seq (Fig 1B and see below) and histology (see below) undertaken. Lung viral titers reached a mean titer of 7.86 $\pm$ SD 0.49 $\log_{10}CCID_{50}$/g on 2 days post infection (dpi), falling to a mean of 4.04 $\pm$ SD 0.8 $\log_{10}CCID_{50}$/g at 5 dpi, and by 10 dpi, 3 of 4 BA.1 infected mice showed no detectable virus titers (Fig 1A). Viral read counts from lungs of BA.1 infected mice, obtained from RNA-Seq data, showed a similar trend, with no significant viral reads detected at 30 and 66 dpi (Fig 1B). No significant read counts were obtained from mice inoculated with UV-inactivated virus (Fig 1B). Although different lung lobes were used for viral titrations and for RNA-Seq, there was a highly significant correlation between these two measures of viral load (S1A Fig in S1 File).

When compared with infection with an original strain isolate (and using the same assay systems) [55,64], infectious viral titers for BA.1 were comparable at 2 dpi, but were $\approx$ 2 logs lower at 5 dpi (Fig 1A). Again using the same assay systems, mean viral reads counts (Fig 1B), representing genomic and subgenomic viral RNA, were $\approx$ 2 logs higher in original strain infected K18-hACE lungs at 2 dpi [55], and did not diminish by more than 0.75 logs by 4 and 7 dpi [55,73]. Mean lung viral genome copy numbers [66] in K18-hACE2 mice were also $\approx$ 6 fold (0.8 log) lower after BA.1 infection than after infection with an original strain isolate (SARS-CoV-2$_{QLD02}$) (Fig 1C). Viral replication and/or transcription was thus lower in BA.1 vs. original strain infected mice, and diminished more rapidly.

As reported previously [54,74,75], most mice in the BA.1/K18-hACE2 model do not develop a brain infection. Brain infection is associated with >20% weight loss and an ensuing ethical requirement for euthanasia (S1B Fig in S1 File). After infection of K18-hACE2 mice with original strain isolates, mice succumb to brain infections between 4 and 7 dpi [54,64]. This BA.1 model thus provides a significant lung infection, and permits analysis of long-term responses to SARS-CoV-2.

### BA.1 infected bronchial epithelial cells and pneumocytes in K18-hACE2 mouse lungs

Immunohistochemistry (IHC) of lungs taken 2 dpi (peak viral load), using an anti-spike monoclonal antibody [60], illustrated clear staining of bronchial epithelial cells (Fig 1D) and cells surrounding alveoli that have a morphology consistent with pneumocytes (Fig 1E). The BA.1/K18-hACE2 model thus recapitulates the main cell tropisms seen in human lung tissues, with infection of bronchial and alveolar epithelial cells by omicron variants well described [76,77].

### BA.1 induced histopathological lesions in lungs resolved by 66 dpi

Hematoxylin and eosin (H&E) stained sections of lung tissues from BA.1 infected K18-hACE2 mice illustrated a series of histopathological features (Figs 2 and S2 in S1 File) previously identified in SARS-CoV-2 mouse models [78], although they appeared less severe than those seen after infection of K18-hACE2 mice with original strain isolates, consistent with previous reports [74,75]. For instance, in the latter, pronounced sloughing of the bronchial epithelium and fulminant occlusions of bronchi (with serum and red blood cells) were observed [79,80].

In BA.1 infected mice (as in other models [59,78]), focal areas of leucocyte infiltration were clearly evident, characterized by dense clusters of dark-purple staining leukocyte nuclei;

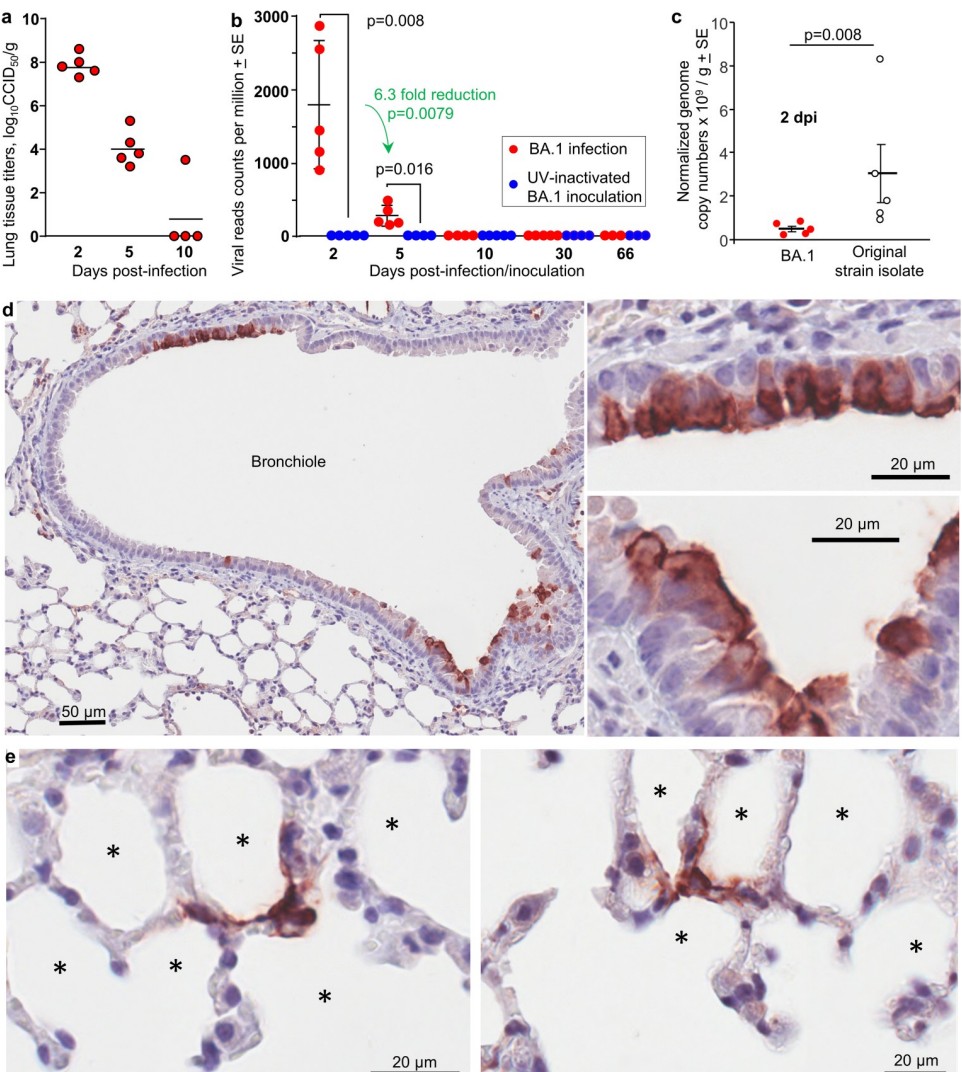

**Fig 1. Viral loads and immunohistochemistry of viral antigen in lungs of BA.1 infected K18-hACE2 mice. a** Lung tissue titers at the indicated days post infection with BA.1 (limit of detection per mouse ≈ 2 $\log_{10}CCID_{50}$/g). **b** Viral read counts per million reads, determined by RNA-Seq for K18-hACE2 mice infected with BA.1 or inoculated with UV-inactivated BA.1 virus. Statistics by Kolmogorov Smirnov exact tests. **c** Viral genome copy number determined by RTqPCR for K18-hACE2 mice infected with BA.1 or an original strain isolated SARS-CoV-2$_{QLD02}$. Statistics by Kolmogorov-Smirnov exact test. **d** Immunohistochemistry of BA.1-infected lungs at 2 dpi, stained with an anti-spike monoclonal antibody. Staining of bronchial epithelia cells (dark brown) can be seen top left and bottom right (left image), with enlargements of these two areas shown in the right 2 images. **e** As for d showing staining of cells that surround alveoli with morphology consistent with that of pneumocytes (alveolar epithelial cells). * alveolar air sacs.

although the infiltrates were more diffuse at 2 dpi (Fig 2A, green dashed lines). These areas were measured manually using the Aperio ImageScope "Pen Tool" (Fig 2A, green dashed lines), providing quantitation of leukocyte infiltration. Significantly higher levels of infiltration were seen for BA.1-infected versus UV-inactivated BA.1-inoculated mice at all time-points except 66 dpi, with infiltrates also substantially reduced by 30 dpi (Fig 2B) (Aperio Positive Pixel Count assessment of leukocyte infiltrates also showed significantly higher infiltrate levels at 5 dpi; S1c Fig in S1 File). Lung consolidation with resulting loss of white space, primarily as a result of diminished alveolar air sack volumes, was also clearly seen at 5 dpi (Fig 2C).

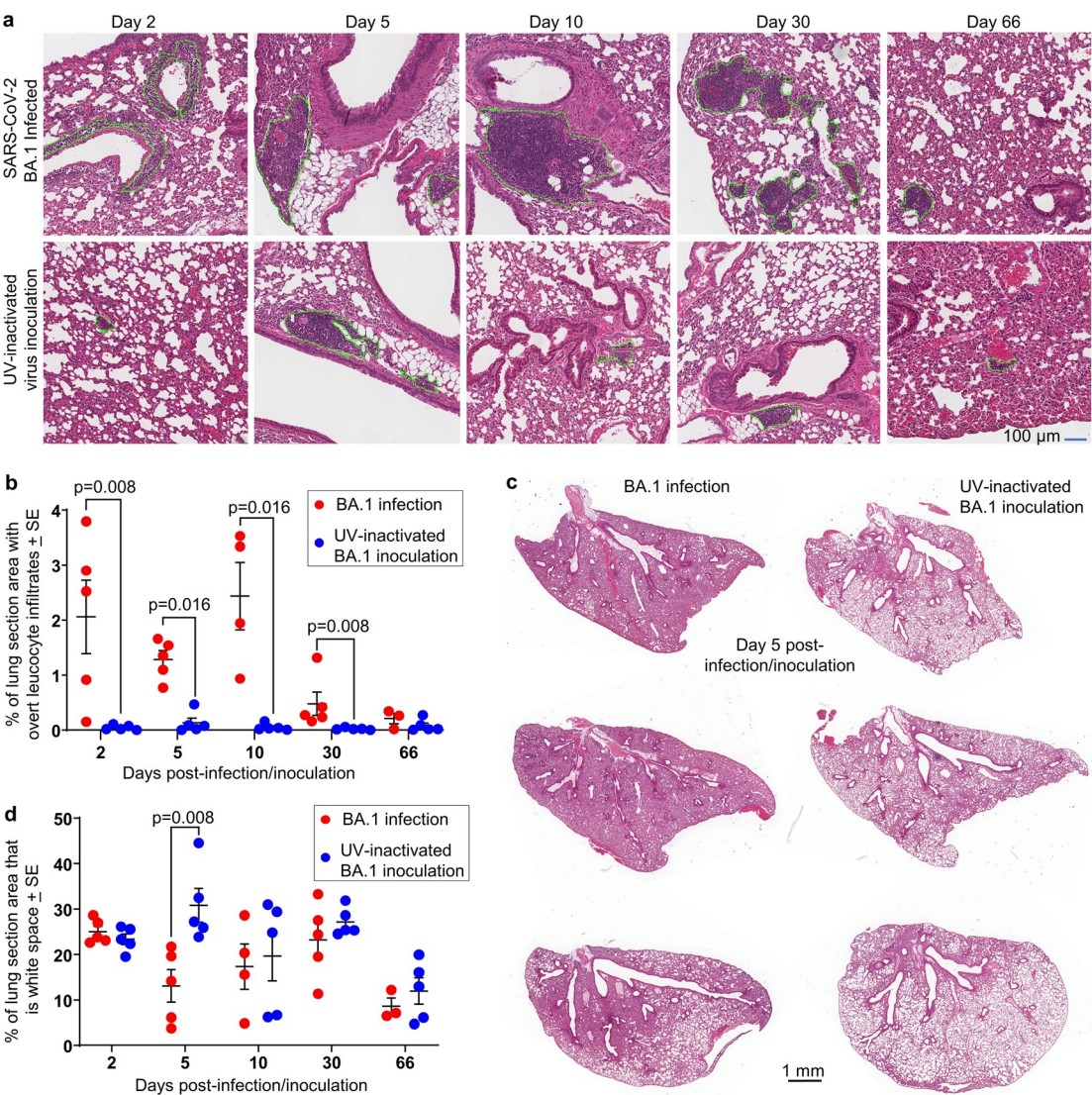

**Fig 2. Histology of lungs after BA.1 infection of K18-hACE2 mice. a** H&E staining of lung tissue sections at the indicated days after infection/inoculation. Green dotted lines encircle overt foci of cellular infiltrates. **b** Quantitation of cellular infiltrates. All cellular infiltrates were marked as in "a" and the summed infiltrate areas expressed as a percentage of the lung section area for each mouse. Statistics by Kolmogorov Smirnov exact tests. (For BA.1 infected mice, data for 2, 5 and 10 dpi are not significantly different, while 30 dpi shows a significant reduction from 5 and 10 dpi by t tests, p = 0.0.019 and p = 0.039, respectively). **c** Whole H&E stained lung sections from day 5 after infection/inoculation, illustrating reduced white space in lungs from infected mice. **d** QuPath digital analysis of white space in H&E stained whole lung sections such as those shown in "c". Statistics by t test.

Significant loss of white space at 5 dpi was confirmed by image analysis of scanned lung sections using QuPath (Fig 2D). Minor intrusion of red blood cells and serum into bronchi, and some alveolar edema were also occasionally observed in BA.1-infected mice (S2 Fig in S1 File). H&E images from PBS and naïve control lungs are shown in S3 Fig in S1 File.

In summary, lungs of BA.1 infected K18-hACE2 mice displayed some typical acute histopathological features described previously for SARS-CoV-2 infections, although less severe than after infection with original strain isolates. At 30 dpi BA.1 infection-associated histopathology was clearly resolving, and by 66 dpi had largely resolved.

## Cell types associated with resolution of inflammation were evident on 5 and 10 dpi

BA.1 infected lungs and UV BA.1 inoculated lungs were compared at each time point (2, 5, 10, 30 and 66 dpi) by RNA-Seq, using DESeq2 [81]. Full gene lists and bioinformatic analyses are provided for 2 dpi (S1 Table), 5 dpi (S2 Table), 10 dpi (S3 Table), 30 dpi (S4 Table), and 66 dpi (S5 Table). RTqPCR was undertaken for 3 genes and two time points to validate the RNA-Seq data, with highly significant concordance emerging for the two methods (S4a Fig in S1 File). We have previously reported high levels of concordance between these two methods in other studies [61,69].

Ingenuity Pathway analysis (IPA) Diseases or Functions annotations for Activation of leukocytes, Chemotaxis of leukocytes, Leukocyte migration and Quantity of leukocytes showed a progressive decline over time, with a slight elevation 10 dpi (Fig 3A), consistent with Fig 2B. To gain insights into the type of cellular infiltrates present in the lungs over time post infection with BA.1, cellular deconvolution (SpacialDecon) analysis using full gene sets (S1–S5 Tables) were used together with the gene expression matrices from the "Mouse Cell Atlas (MCA) Lung Cell expression matrix" and the "NanoString (NS) Immune Cell Family expression matrix". As might be expected [82,83], a series of innate mononuclear leukocyte populations were identified at 5 dpi, which included macrophages, monocytes, dendritic cells, and NK cells, as well as gamma delta T cells [84] (Fig 3B). Increased abundance of nuocytes, ILC2 cells, was also identified on both 5 and 10 dpi (Fig 3B and 3C, nuocytes). These innate effector leukocytes mediate type-2 immunity [85] and are involved in the initiation of inflammation, but also have key roles in damage control, homeostasis and repair [86,87]. Lower frequencies of ILC2 cells have been associated with severe COVID-19 [88]. An increase abundance score for pericytes was also identified in infected lungs at 5 dpi (but not other times) (S4b Fig in S1 File). Pericytes express ACE2 and binding of spike protein has been shown *in vitro* to induce profound transcriptional changes in these cells [89,90], which may explain this result.

At 10 dpi, in addition to nuocytes, significantly increased abundance of three more cell types associated with repair and return to homeostasis was evident (Fig 3C, blue text). Inducible T regulatory cells (T regs) characterised by low neurophilin expression (Nrp-1$^{lo}$) [91] were more abundant in infected lungs. T regs secrete *inter alia* IL-10 and are generally protective [92], not just by dampening proinflammatory responses, but also via prevention of fibrosis and maintenance of tissue homeostasis [93]. Interstitial macrophages were also elevated in infected lungs, with these cells having an inherently anti-inflammatory phenotype, secreting *inter alia* IL-10 (USR z score 1.69, q = 1.39E-29, S3 Table) and supporting T regs in the lung [94]. An Ear2+ alveolar macrophage population was also identified, with Nra4a1+, Ear2+ macrophages recently found to have a novel reparative phenotype [95]. Although perhaps proinflammatory, CD8 T cells were also more abundant at 10 dpi (Fig 3C), with CD8 T cells recently identified as key to protection against SARS-CoV-2 [96].

## Cytokine signatures in BA.1 infected lungs and their resolution by 30–66 dpi

Differentially expressed genes (DEGs, q<0.05) (S1–S5 Tables) were analyzed by IPA, with cytokine up-stream regulator (USR) z scores and p values illustrated by heat maps (Fig 4A). The acute cytokine responses were broadly comparable to those published previously for a K18-hACE2 mouse model of SARS-CoV-2 infection with an original strain isolate; specifically, hCoV-19/Australia/QLD02/2020 with the same viral dose of $5 \times 10^4$ CCID$_{50}$, the same route and volume of inoculation, and also female mice (NCBI SRA Bioproject PRJNA767499) [55]. However, when cytokine USR z scores at 2 dpi from BA.1 infection (Fig 4A) were compared

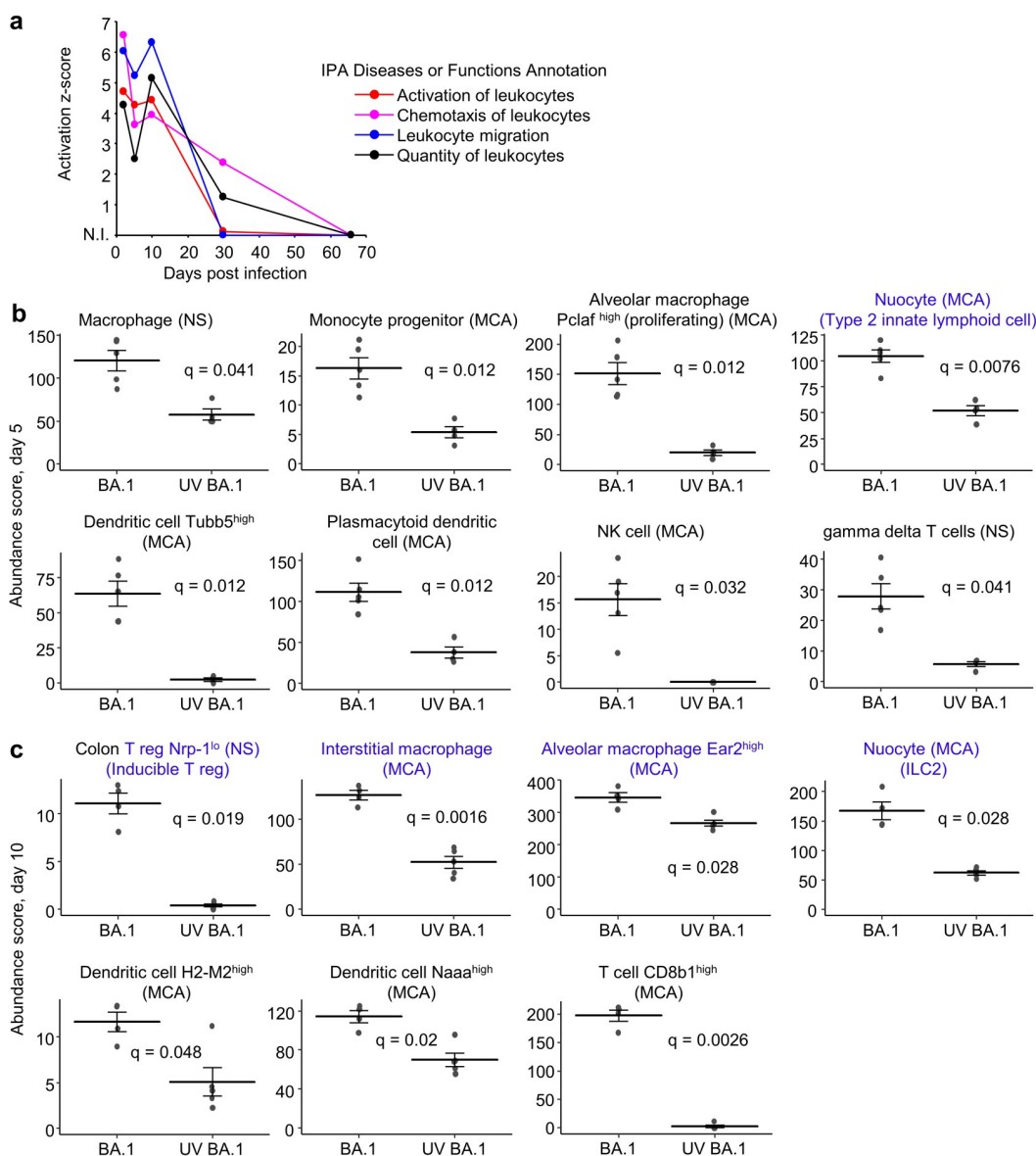

**Fig 3. Cellular infiltrates and types identified by RNA-Seq of lungs for 2–30 dpi. a** Z scores for the indicated annotations from Ingenuity Pathway analysis (IPA) Diseases or Function analysis plotted against dpi. (N.I. not identified; for 66 dpi there were insufficient DEGs for IPA analysis). **b** Using cellular deconvolution (SpatialDecon) and the gene expression matrices from the "Mouse Cell Atlas (MCA) Lung Cell expression matrix" and the "NanoString (NS) Immune Cell Family expression matrix", cell types whose abundance scores were significantly different on day 5 were identified (q values provided). Blue text indicates cell type associated with inflammation resolution. **c** As for b but for day 10. Blue text indicates cell type associated with inflammation resolution. We assume "colon" T reg are well annotated in the NS matrices (and are thus readily identified), and represent tissue T regs, rather than implying that colonic T regs have migrated to the lungs.

with those from the aforementioned infection with the original strain isolate [55], small differences in z scores (between ≈1 and -1) emerged (Fig 4B). A series of USRs associated with less severe disease had slightly higher z score in BA.1 infected mice; these included EPO (erythropoietin) [97], IL-15 [98], and increased type I interferon responses [18,50] (Fig 4B). The z score was also higher for IL-10, with IL-10 elevated in severe COVID-19, but also able to suppress hyper-cytokinemia [19–21]. Elevated IL-4, IL-9 and IL-13 have been associated with

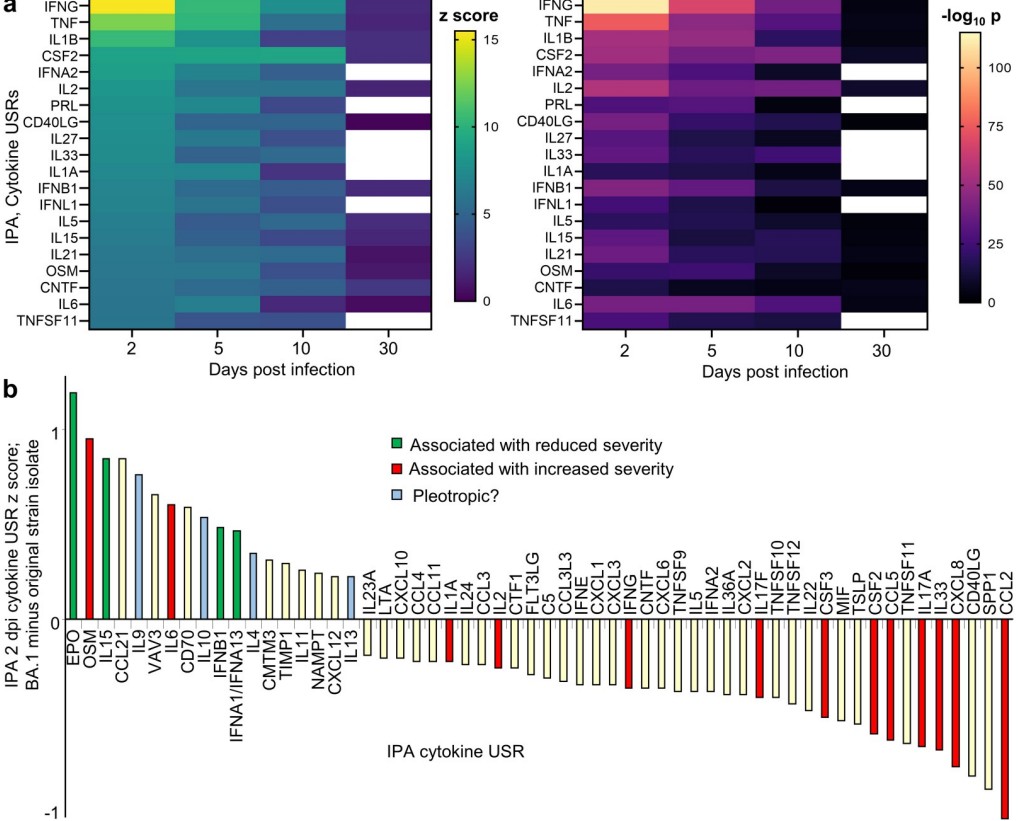

**Fig 4. Inflammatory pathways identified by RNA-Seq of lungs for 2–30 dpi. a** Heat maps of cytokine UpStream Regulator (USR) z scores and significance (p values, all p<0.05) determined by IPA. White boxes means no cytokine USR annotation was returned for 30 dpi by IPA. RNA-Seq identified too few DEGs on 66 dpi for IPA analysis. **b** IPA cytokine USRs from BA.1 infected K18-hACE2 mice (2 dpi) were compared with those previously reported for the more severe infection of K18-hACE2 mice by an original strain isolate [55]. Differences in cytokine z scores for 2 dpi (with a cutoff filter of >0.2 and < -0.2) are plotted. BA.1 infected lungs show slightly increased z scores for some cytokine signatures associated with reduced disease severity (green) and slightly reduced z scores for some cytokine signatures associated with increased disease severity (red).

more severe disease [22,23,99], although IL-9, IL-13 (and IL-5) are produced by type 2 innate lymphoid cells (ILC2 cells) [100] (see below), and increased IL-4 and IL-13 signatures may (in this context) reflect a relatively more balanced Th1/Th2 response [26,27]. Elevated OSM and IL-6 are also associated with more severe disease [18,101]. Another group of cytokines/chemokines associated with increased disease severity (IL-2, IFNG, IL-17, CSF2, CCL5, CXCL8, CCL2 [11,102,103] showed slightly lower z scores for BA.1 infected lungs (Fig 4B). IL-33 may drive fibrosis [104] and also showed a lower USR z score (Fig 4B).

Importantly, the cytokine response signatures had substantially abated by 30 dpi (Fig 4A), with the small number of DEGs on day 66 insufficient for meaningful pathway analysis. In addition, few, if any, upregulated inflammatory DEGs were identified at 66 dpi (see below).

## Transcriptional responses 66 dpi reflected adaptive immunity and resolution

The RNA-Seq data sets and bioinformatic analyses comparing lungs from BA.1 infected mice 66 dpi with mice 66 days after inoculation with UV-inactivated virus (S5 Table) are summarized in Fig 5. Only 29 DEGs were identified, of which 14 were immunoglobulin genes. This

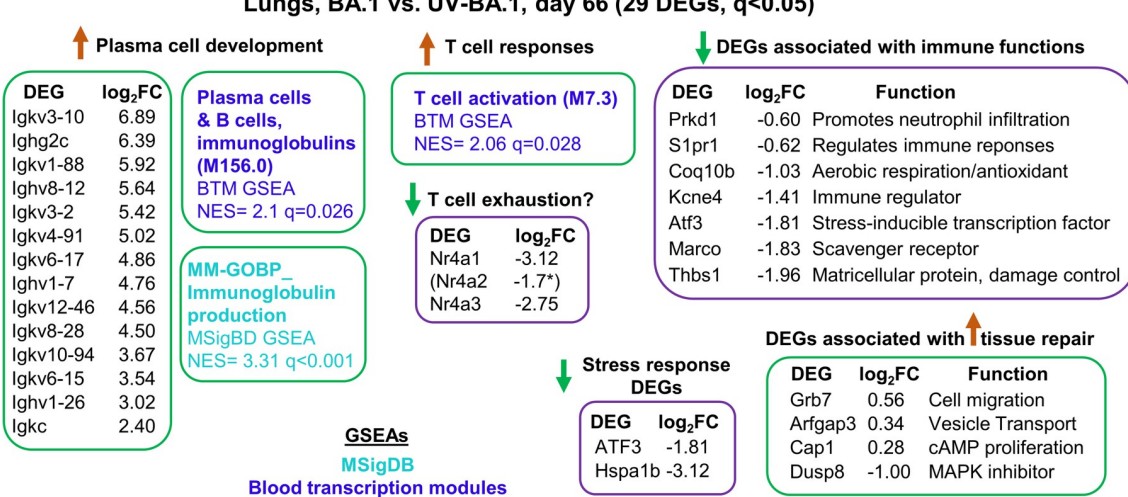

**Fig 5. DEGs identified for 66 dpi.** RNA-Seq data for day 66 dpi, with DEGs clustered by broad functions. The whole gene list was also interrogated by GSEAs using gene sets from MSigDB and Blood transcription modules.

represents an insufficient number of DEGs for meaningful pathway analyses. DEGs were thus clustered into several groupings based on function (Fig 5, described below). A plot of the first two principle components of normalized Variance Stabilizing Transformation (VST) counts from all experimental groups indicated that the relatively small number of DEGs for day 66 was due to limited separation between BA.5 infected and UV control groups (S5 Fig in S1 File).

Most of the upregulated DEGs in lungs at 66 dpi were immunoglobulin genes (Fig 5) consistent with a more rigorous development of an adaptive immune response in BA.1 infected mice compared with mice inoculated with UV-inactivated virus. GSEAs also illustrate the development of T cell responses, with down-regulation of Nr4a family members potentially suggesting avoidance of T cell exhaustion [105]. However, the three Nr4a genes have a number of roles in T cell biology [106], and also have a diverse set of other activities in *inter alia* macrophages [95,107] and epithelial cells [69].

The three upregulated, non-Ig genes, can be associated with wound healing/tissue repair; Grb7 (Growth factor receptor-bound protein 7) [108], Arfgap3 (ADP-ribosylation factor GTPase-activating protein 3) [109,110], and Cap1 (cyclase-associated protein 1) [111]. In addition, Dusp8 (Dual specificity phosphatase 8) was down-regulated, with Dusp8 protein involved in negative regulation of MAP kinase superfamily members; thus Dusp down-regulation would promote cellular proliferation and differentiation [112].

Many of the remaining down-regulated genes have roles in various immune activities (Fig 5): Prkd1 (Serine/threonine-protein kinase D1), involved in *inter alia* promoting neutrophil infiltration [113]; S1pr1 (Sphingosine-1-phosphate receptor 1), essential for immune cell trafficking [114]; Coq10b, required for coenzyme Q function [115,116]; Kcne4 (Potassium voltage-gated channel subfamily E member 4), an immune regulator via modulation of Kv1.3 [117]; Atf3, a stress-inducible transcription factor [118]; Marco, a scavenger receptor [119]; and Thbs1 (thrombospondin), a matricellular protein involved in lung inflammation resolution and repair [120]. Two stress-inducible genes, Atf3 and Hspa1b (a heat shock protein 70 member) were down-regulated, with Atf3 and Hsp70 upregulated in a large range of stress responses [69,118].

In summary, at 66 dpi there were very few DEGs, with no indication of ongoing inflammation or tissue dysregulation, with identification of only a small number of upregulated genes that could be associated with wound healing processes.

## Genes whose expression positively correlated with dpi were associated with ciliated epithelial cells and protein synthesis

Using the complete gene sets (S1–S5 Tables), genes were identified whose expression showed significant positive correlation with dpi (Pearson correlations, FDR corrected p, q<0.05). Genes whose expression also showed significant positive correlation with days post inoculation of UV-inactivated virus were excluded from the analysis, thereby retaining only genes associated with active infection. Immunoglobulin genes were also removed as the development of immune responses is understood and established. The process identified 2786 genes (S6 Table).

Enrichr and IPA analyses of the 2786 genes indicated a significant association with ciliated epithelial cells and protein synthesis, with 59 of the 2786 representing ribosomal protein genes (Fig 6A and S6 Table). Ciliated epithelial cells are a known target of SARS-CoV-2 infection [64,121], with omicron variants showing a higher propensity to infect these cells [76]. The increase in ciliated epithelial cell genes thus indicates growth/recovery and differentiation of these cells over the 2–66 days of the experiment. The increase in signatures associated with protein translation during this period is consistent with the development of adaptive immune responses and ongoing tissue repair processes.

## Genes whose expression negatively correlated with dpi indicated inflammation resolution

Using the complete gene sets, genes were identified whose expression showed significant negative correlation with dpi (Pearson correlations, q<0.05). Genes whose expression also showed significant negative correlation with days post inoculation of UV-inactivated virus were excluded from the analysis, thereby retaining only genes associated with active infection. The process identified 1996 genes (S6 Table).

IPA analyses of the 1996 genes identified reductions in cell movement/migration (Fig 6B), consistent with Figs 2A, 2B and 3A. A series of cytokine USRs were also down-regulated (S6 Table), with the top cytokine USRs (by negative z score) representing cytokines previously identified as being robustly associated with COVID-19 disease severity [11]. The latter was supported by identification of "Pathogen Induced Cytokine Storm Signaling Pathway" as a top IPA Canonical pathway annotation with a negative z score (Fig 6B). Also informative were decreasing neutrophil degranulation and fibrosis signatures (Fig 6B); both phenomena have been associated with long COVID [122,123].

Negative correlations between expression and dpi are plotted for five of the 1996 genes (Fig 6C). These genes were selected as their expression levels have been positively correlated with COVID-19 disease severity and/or they have been proposed as potential targets for prognosis or treatment; SerpinE1 [124], Cxcl10 [125], Nfkb1 [126], IL6 [127], and Hspa1b (a Hsp70 family member) [128,129].

## Cell types whose abundance scores increased with dpi indicated lung regeneration

Cell types whose abundance scores (determined by cellular deconvolution) showed a significant positive correlation (Pearsons q<0.05) with dpi were identified. The top scoring cell type was ciliated cells (Fig 6D); consistent with Fig 6A. The analysis also identified significant

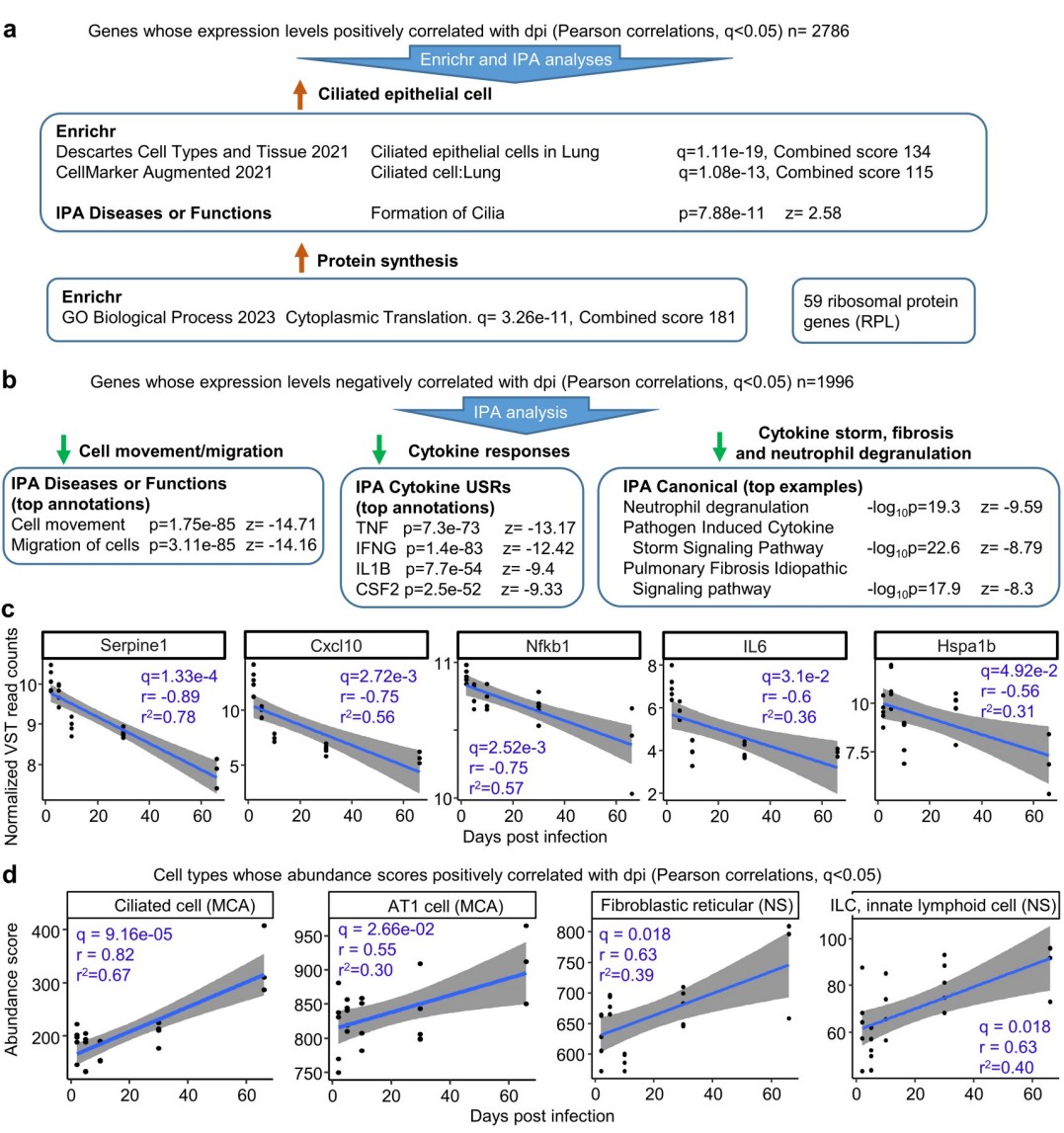

**Fig 6. Genes whose expression correlated with dpi. a** Using the whole gene lists, genes whose expression showed a statically significant positive correlation with dpi were identified. After removing genes that also positively correlated with dpi in the UV-inactivated BA.1 inoculated control groups, 2786 genes remained. The 2786 genes were analyzed by Enrichr and IPA (for the latter the r value was entered in the log2 expression column). Significant results clustered into two categories 'Ciliated epithelial cell' and 'Protein synthesis'. Of the 2786 genes, 59 were RPL (ribosomal protein) genes. **b** Using the whole gene lists, genes whose expression showed a statically significant negative correlation with dpi were identified. After removing genes that also negatively correlated with dpi in the UV control groups, 1996 genes remained. The 1996 genes were analyzed by IPA as above, with some top annotations shown. **c** Normalized expression data (VST - Variance Stabilizing Transformation) for five of the 1996 genes are plotted against dpi, with linear regression lines and 95% confidence intervals. Pearson correlation coefficients (r), coefficients of determination ($r^2$) and corrected significances, FDR (q), are provided for each gene. **d** Using cellular deconvolution (as in Fig 3B and 3C), cell types whose abundance scores significantly positive correlated with dpi were identified. Linear regression lines and 95% confidence intervals, and Pearson correlation coefficients (r), coefficients of determination ($r^2$) and corrected significances, FDR (q) are provided. (Full data sets are provided in S6 Table).

increases in AT1 cells, which likely reflects the AT2 to AT1 transition during alveolar epithelial cell regeneration seen after SARS-CoV-2 alveolar damage in mouse models [130]. Increasing abundance scores for reticular fibroblasts, are also consistent with successful lung repair

[51,131], with no indication of significant fibrosis. Increasing numbers of innate lymphoid cells (ILC) likely also reflect the process of returning the tissue to homeostasis [132,133].

## Discussion

We describe herein the transcriptional profile of BA.1 omicron lung infections in K18-hACE2 mice, and the post-infection recovery up to 66 dpi. The expression of hACE2 via the keratin 18 promoter in the K18-hACE2 mice did not lead to overtly aberrant cell tropism in the lungs, with infection of bronchial and alveolar epithelia cells by omicron variants (Fig 1D and 1E) also described in settings where hACE2 is not expressed as a transgene [76,77]. Although there are some differences in methodologies, the overall consensus is that, when compared to infection with original strain isolates, BA.1 viral loads in lungs are lower and/or are more rapidly cleared in rodent models [74,75,134–137]. The BA.1 data presented herein (Fig 1A–1C) supports this contention when compared with previously published data on infection of K18-hACE2 mice with original strain isolates [55,64,73].

Herein we show that only a small number of DEGs were identified at 66 dpi with BA.1, arguing that the infection had been resolved and the tissue had largely returned to normal at this time. UV-inactivated virus was used as a control to exclude responses to viral inoculation and restrict the analyses to effects arising from replicating virus. Features often associated with post-acute or long-COVID were not identified, for instance, there were no indications of prolonged infection [138], persistent inflammation [139], impaired lung regeneration [140], fibrosis [123], prolonged T cell activation [141], Th17 bias [142], chronic neutrophil activation [122] or clotting abnormalities [8]. Only 8 mice were analyzed at > 30 dpi, which is arguably too low to identify a potential subset of individuals with post-acute or long-COVID [143,144]. Nevertheless, the overall picture that emerges for the BA.1 infection of K18-hACE2 is one of an attenuated, rapidly resolving, sequelae-free, infection. Importantly, lower viral loads and more rapid viral clearance are also associated in COVID-19 patients with less severe disease [35] and/or subclinical infections [46,47].

An inherent limitation with studies using K18-hACE2 mice is the level to which these animals recapitulate human infection and disease. For instance, fulminant lethal brain infections, usually seen 4–7 days after infection with original strain isolates, is not a feature of human disease [54,145,146]. However, such lethality is rare after BA.1 infections (S1b Fig in S1 File), thereby permitting the investigation of long-term responses presented herein. We have also shown that overlap between DEGs identified by RNA-Seq analysis of infected lung tissues from humans vs. K18-hACE2 mice is generally quite low; however, a high level of correlation emerged between inflammatory pathways [55]. Key clinical parameters used to evaluate disease severity in COVID-19 patients (e.g. oxygen saturation, respiration rate, chest imaging, full blood works) are also not readily available for mice in BSL3/PC3 settings. In addition, although lung samples are easily obtained from infected mice, they are not so readily available from COVID-19 patients. Finally, the plethora of different virus isolates, virus doses, rodent models (including age and gender), and analytical techniques, complicates detailed comparisons across studies [147–150]. For instance, although RNA-Seq data suggests increased pathogenicity of delta variants over ancestral (original) strain isolates in K18-hACE2 mice [148], numerous methodological differences complicate integration of such data into compelling comparisons with our study. Finally, it remains unclear to what extent inbred syngeneic mouse strains (with specific genetic backgrounds [63]) can faithfully recapitulate the activity of key genetic traits (e.g. HLA-DQA2) that underpin rapid SARS-CoV-2 clearance in humans [40].

The concept of "protective inflammation" was recently described for COVID-19 [56] and clearly applies to many other viral infections where subclinical or mild infections are also well

documented [151]. Deleterious or pathological inflammatory disease processes tend to be the focus of research, with perhaps less attention paid to characterisation of inflammatory responses that are resolved and lead to subclinical or mild disease and sequelae-free outcomes [152,153]. Pathological versus protective responses are likely to have distinct gene expression profiles, patterns, magnitudes and sequence of events [57,58], with the current study describing the responses associated with resolution of lung infection and inflammation. Importantly, at 10 dpi a series of cell types associated with damage control, suppression of inflammation, homeostasis and repair showed increased abundance scores (determined using transcription data from lung tissues and cellular deconvolution). Specifically these cells were, ILC2 cells [86,87] (also seen at 5 dpi), inducible T regs [92,93], interstitial macrophages [94] and Ear2 + macrophages [95]. Restoration of ciliated cells, regeneration of alveolar epithelial cell with AT2 to AT1 transition [130], and increases in fibroblast activity without fibrosis [51], also appear to characterize lung recovery in this model. Clearly, prompt clearance of virus [46,50] (rather than viral persistence [138,154]) and development of protective adaptive immune responses [49] are also likely to be key to such outcomes. The latter has recently been shown to include CD8 T cells [96], which may be represented by a population of CD8b1[high] cells which reached significance by 10 dpi in the BA.1/K18-hACE2 model.

Defining key aspects of "protective inflammation" remains a challenge for human populations, as individuals with subclinical or mild disease are generally difficult to identify and recruit promptly. A better understanding of what characterizes rapid, sequelae-free resolution of inflammation, would clearly facilitate development of interventions for COVID-19 associated ARDS, with animal models providing ready access to lung tissues to help unravel the complex mechanisms that underpin such processes.

## Supporting information

**S1 File. Supplementary figures 1 to 5.**
(PDF)

**S1 Table. Differential expression and pathway analysis for 2 dpi.**
(XLSX)

**S2 Table. Differential expression and pathway analysis for 5 dpi.**
(XLSX)

**S3 Table. Differential expression and pathway analysis for 10 dpi.**
(XLSX)

**S4 Table. Differential expression and pathway analysis for 30 dpi.**
(XLSX)

**S5 Table. Differential expression and pathway analysis for 66 dpi.**
(XLSX)

**S6 Table. Gene expression vs dpi.**
(XLSX)

## Acknowledgments

The authors thank the following staff at QIMR Berghofer MRI; Dr I. Anraku for management of the PC3 facility, Dr Viviana Lutzky for proof reading, Ashwini R Potadar and Crystal Chang for histology services, the animal house staff for mouse breeding and agistment, and Dr. Gunter Hartel for assistance with statistics.

## Author Contributions

**Conceptualization:** Cameron R. Bishop, Daniel J. Rawle, Andreas Suhrbier.

**Data curation:** Cameron R. Bishop, Andreas Suhrbier.

**Funding acquisition:** Daniel J. Rawle, Andreas Suhrbier.

**Investigation:** Agnes Carolin, Kexin Yan, Bing Tang, Wilson Nguyen.

**Methodology:** Cameron R. Bishop, Daniel J. Rawle, Andreas Suhrbier.

**Supervision:** Cameron R. Bishop, Wilson Nguyen, Daniel J. Rawle, Andreas Suhrbier.

**Visualization:** Agnes Carolin, Kexin Yan, Cameron R. Bishop, Andreas Suhrbier.

**Writing – original draft:** Andreas Suhrbier.

**Writing – review & editing:** Agnes Carolin, Cameron R. Bishop, Daniel J. Rawle.

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
