## [Decision Letter · Decision Letter 0]

10 Jun 2024

PONE-D-24-09241Tracking inflammation resolution signatures in lungs after SARS-CoV-2 omicron BA.1 infection of K18-hACE2 micePLOS ONE

Dear Dr. Suhrbier,

Thank you for submitting your manuscript to PLOS ONE. After careful consideration, we feel that it has merit but does not fully meet PLOS ONE’s publication criteria as it currently stands. Therefore, we invite you to submit a revised version of the manuscript that addresses the points raised during the review process.

Your manuscript has been reviewed by experts in the field and by editors. Please revise your document accordingly. Specifically, clarification is needed on the reuse of data in some figures, as raised by Reviewer 1. Additionally, a clear explanation/details of some methods used and improvements to the discussion section are needed as suggested. Please address each comment with revisions to your manuscript or provide an explanation if necessary.

We look forward to receiving your revised manuscript.

Kind regards,

Engin Berber, D.V.M., Ph.D.

Academic Editor

PLOS ONE

Journal Requirements:

2. To comply with PLOS ONE submissions requirements, in your Methods section, please provide additional information regarding the experiments involving animals and ensure you have included details on (1) methods of anesthesia and/or analgesia, and (2) efforts to alleviate suffering.

4. Thank you for stating the following financial disclosure: "The authors thank the Brazil Family Foundation (and others) for their generous philanthropic donations that helped set up the PC3 (BSL3) SARS-CoV-2 research facility at QIMR Berghofer MRI, as well as ongoing research into SARS-CoV-2, COVID-19 and long-COVID.  A.S. is supported by the National Health and Medical Research Council (NHMRC) of Australia (Investigator grant APP1173880).  "

Reviewers' comments:

Reviewer's Responses to Questions

**Comments to the Author**

1. Is the manuscript technically sound, and do the data support the conclusions?

Reviewer #1: Partly

Reviewer #2: Partly

2. Has the statistical analysis been performed appropriately and rigorously? 

Reviewer #1: Yes

Reviewer #2: Yes

3. Have the authors made all data underlying the findings in their manuscript fully available?

Reviewer #1: Yes

Reviewer #2: Yes

4. Is the manuscript presented in an intelligible fashion and written in standard English?

Reviewer #1: Yes

Reviewer #2: Yes

5. Review Comments to the Author

Reviewer #1: Manuscript: PONE-D-24-09241.pdf

The research investigates the dynamics of SARS-CoV-2 omicron BA.1 infection in K18-hACE2 mice, focusing on lung infection and its subsequent clearance. Mice were intranasally inoculated with either live BA.1 virus or UV-inactivated BA.1 virus. Lung tissue was harvested at various days post-infection (dpi) to assess viral titers, RNA levels, and histopathology. Initial results indicated that by 2 dpi, lung viral titers peaked and then significantly declined by 10 dpi, where most mice showed no detectable virus. Viral RNA levels and immunohistochemistry confirmed active infection initially but showed significant reduction in viral load over time. Histological analysis of the lungs showed that infection-induced damage was less severe compared to infections with original strain isolates, with most histopathological features resolving by 66 dpi. The study highlights the concept of “protective inflammation” which is the effectiveness of inflammation in clearing an infection leading to little to no disease, and highlights the fact that most publications focus on the outcomes of severe disease. This is one of the strengths of this manuscript. There are two major concerns with the manuscript in its current state. The clear discussion of protective inflammation is only brought up in the discussion, this concept should start at the abstract and be present throughout. There are minor mentions within the abstract and introduction but it is not consistent enough to have an impact. The second concern is the level of comparisons to prior published work. It is stated and referenced but a more thorough, clear and concise application needs to be done, more described below.

Major concerns

It appears that data is reused within the manuscript at several points within the manuscript. The re-use is referenced (line 219) “Regraphed from Bishop et al 2022”. If this is permitted as a comparison additional experimental detail is needed to allow the comparison, such as how were the two virus stocks titers calibrated? Where the original strain and BA.1 strain tittered by the same method?

Overall a table should include what data has been used previously and list critical information for comparison. Currently it is mentioned in the text and several references provided, but as one example there are multiple strains of mice described in this manuscript as well the ones referenced. In this manuscript both heterozygous and homozygous K18-hACE2 are described, in the referenced manuscripts there are also mACE2 promoter-hACE2 mice described. Where is the data coming from, what are the exact viruses used. A reader needs to be able to track and compare across the multiple manuscripts.

Minor concerns-

Several sections lack detail or definitions. For example CCID50 is not defined.

Reviewer #2: Dear editor,

The manuscript “Tracking inflammation resolution signatures in lungs after SARS-CoV-2 omicron BA.1 infection of K18-hACE2 mice” by Agnes Carolin et al. aims at analyzing the lung response to infection with SARS-CoV-2 Omicron BA.1 in a K-18 hACE2 transgenic mouse model. The authors analyze RNA seq data from lungs of mice that were either infected with a replicating virus or exposed to a UV irradiated virus as a control. As the BA.1 variant is significantly less pathogenic to both humans and to hACE-2 Tg mice, they are mainly focused on the immune response that leads to resolution.

The authors applied multiple analyses to unravel the pathways involved in resolution of infection and the manuscript is overall coherent and valuable. However, several issues need to be considered and addressed to satisfy the standards for publication in PLOS one, as listed below:

Major issues:

1. The authors fail to properly reference other publications in the field, such as the analyses of RNA seq of K-18 mice infected with other variants. The differences between the pathways at the early period after infection should be discussed. Indeed, the previous variants were lethal, and pathways were associated with pathogenicity, overt immune response and so on, so pathways associated with recovery, presented in the reviewed manuscript, are unique to Omicron and the differences are important and should be discussed.

2. Analyses of the infected lungs are based on lobes rather than whole lung. Unless the authors demonstrate the homogeneity of infection and of histological changes throughout the lung, taking parts (lobes) of the lungs for different analyses is not reliable.

3. Fig. 1b&c –Viral load data were determined by RNA seq data and are presented as reads per million counts. In b the vertical axis is misleading (1,5,10 than 10 again and then 1E4.) These results are hard to translate to viral load/copy and do not account for genome/antigenome copies and it is recommended to apply real-time RT-PCR to covert the data to genome equivalents. Furthermore, the data in 1c show about 1 log higher copies compared to 1b – whereas the authors indicate that the loads at day 2 post infection are comparable (line 208).

4. The data in graph 1c is from reference #53 and while properly reported as such, should not be part of the results (1c, lines 227-8). These data should be discussed in the discussion.

5. Fig. 1 d-e – while the authors demonstrate a viral load of about 1E8 CCID50/gr lung (about 2E7/ lung) at day 2, the immunohisthology shows at the same day very few stained areas. Unless image of the whole lung/lobe shows widespread viral antigens throughout the lung to account for the high viral load, the data are inconsistent.

6. Fig. 2b and lines 268-281 – The analyses presented in 2b shows inconsistent yet significant effects following infection with the replicating virus (red) up to day 10. This should be discussed.

7. Fig. 4 – The Z score describes only one aspect of the analysis (e.g. activation/inhibition). -Log P value should be added to demonstrate the robustness of the observed responses.

8. Fig. 6b – The data in the three “boxes” seems to be provided in different ways – either p for the two right boxes and -log P value for the left box. It should be corrected such that the values are presented in a similar way.

9. Fig. 6 c&d – R2 should also be provided.

10. RNA seq. data should be validated.

11. Discussion – lines 491-494 – The authors mention that long-COVID features were not identified but with the limited number of mice and the frequency of long-COVID in the population their expectation is unreasonable. The discussion should be corrected.

Minor:

1. Detailed description of RNA isolation and preparation of libraries are missing. Placing organs in RNA later is not sufficient.

2. Line 262 – One lobe is not “total lung”.

publication in PLOS ONE, research articles must satisfy the following criteria:

1. The study presents the results of primary scientific research – yes.

2. Results reported have not been published elsewhere – True.

3. Experiments, statistics, and other analyses are performed to a high technical standard and are described in sufficient detail. – Major corrections are needed.

4. Conclusions are presented in an appropriate fashion and are supported by the data. As in q3 the results should be properly analyzed and presented and then the conclusions should be written.

5. The article is presented in an intelligible fashion and is written in standard English – True.

6. The research meets all applicable standards for the ethics of experimentation and research integrity - Yes

7. The article adheres to appropriate reporting guidelines and community standards for data availability.- Yes.

6. PLOS authors have the option to publish the peer review history of their article (what does this mean?). If published, this will include your full peer review and any attached files.

Reviewer #1: No

Reviewer #2: No

---

## [Author Response · Author response to Decision Letter 0]

2 Aug 2024

Response to reviewers’ comments: PONE-D-24-09241; Tracking inflammation resolution signatures in lungs after SARS-CoV-2 omicron BA.1 infection of K18-hACE2 mice.

Reviewers’ comments are reproduced in italics to facilitate re-review.

Reviewer #1: 

The research investigates the dynamics of SARS-CoV-2 omicron BA.1 infection in K18-hACE2 mice, focusing on lung infection and its subsequent clearance. Mice were intranasally inoculated with either live BA.1 virus or UV-inactivated BA.1 virus. Lung tissue was harvested at various days post-infection (dpi) to assess viral titers, RNA levels, and histopathology. Initial results indicated that by 2 dpi, lung viral titers peaked and then significantly declined by 10 dpi, where most mice showed no detectable virus. Viral RNA levels and immunohistochemistry confirmed active infection initially but showed significant reduction in viral load over time. Histological analysis of the lungs showed that infection-induced damage was less severe compared to infections with original strain isolates, with most histopathological features resolving by 66 dpi. The study highlights the concept of “protective inflammation” which is the effectiveness of inflammation in clearing an infection leading to little to no disease, and highlights the fact that most publications focus on the outcomes of severe disease. This is one of the strengths of this manuscript. There are two major concerns with the manuscript in its current state. 

Comment #1: The clear discussion of protective inflammation is only brought up in the discussion, this concept should start at the abstract and be present throughout. 

Response: We have added “with reference to the concept of protective inflammation” to the last sentence in the Abstract. Protective inflammation is also described in some detail at the end of the Introduction. 

Comment #2: There are minor mentions within the abstract and introduction but it is not consistent enough to have an impact. 

Response: We have added “Protective responses are likely to have distinct gene expression patterns and modulated cellular infiltrates, when compared with inflammatory disease [56, 57]” to the Introduction.

Comment #3: The second concern is the level of comparisons to prior published work. It is stated and referenced but a more thorough, clear and concise application needs to be done, more described below.

Major concerns

It appears that data is reused within the manuscript at several points within the manuscript. The re-use is referenced (line 219) “Regraphed from Bishop et al 2022”. If this is permitted as a comparison additional experimental detail is needed to allow the comparison, such as how were the two virus stocks titers calibrated? Where the original strain and BA.1 strain tittered by the same method? Overall a table should include what data has been used previously and list critical information for comparison. Currently it is mentioned in the text and several references provided, but as one example there are multiple strains of mice described in this manuscript as well the ones referenced. In this manuscript both heterozygous and homozygous K18-hACE2 are described, in the referenced manuscripts there are also mACE2 promoter-hACE2 mice described. Where is the data coming from, what are the exact viruses used. A reader needs to be able to track and compare across the multiple manuscripts.

Response: Also commented on by Reviewer 2. We have removed Fig. 1c from the manuscript entirely as it represents data that has indeed been published previously and methodologies are indeed slightly different. Although such differences perhaps preclude our attempts at direct side by side comparisons, that original strain isolates replicate better than omicron isolates in various rodent models has now been well described by a number of groups using a range of slightly different methodologies. Perhaps by providing such specific detailed data from our own group, we have over-complicated this issue and given the wrong impression that we are trying to provide some kind of exact side-by-side comparison. 

 We have now added a section to the discussion (as recommended by reviewer 2) to make the key point that BA.1 lung infection is generally found to be attenuated (consistent with our data), and have also provided a series of references from various groups using a range of methodologies that support this contention. We have also provided, in the Results section, specific details of the mouse model and virus isolate for the Fig. 4b comparison.

Comment #4: Minor concerns-

Several sections lack detail or definitions. For example CCID50 is not defined.

Response: We have clarified at first use (in Materials and methods) that this is cell culture infectious dose 50%.

Reviewer #2

The manuscript “Tracking inflammation resolution signatures in lungs after SARS-CoV-2 omicron BA.1 infection of K18-hACE2 mice” by Agnes Carolin et al. aims at analyzing the lung response to infection with SARS-CoV-2 Omicron BA.1 in a K-18 hACE2 transgenic mouse model. The authors analyze RNA seq data from lungs of mice that were either infected with a replicating virus or exposed to a UV irradiated virus as a control. As the BA.1 variant is significantly less pathogenic to both humans and to hACE-2 Tg mice, they are mainly focused on the immune response that leads to resolution.

The authors applied multiple analyses to unravel the pathways involved in resolution of infection and the manuscript is overall coherent and valuable. However, several issues need to be considered and addressed to satisfy the standards for publication in PLOS one, as listed below:

Major issues:

Comment #1: 1. The authors fail to properly reference other publications in the field, such as the analyses of RNA seq of K-18 mice infected with other variants. The differences between the pathways at the early period after infection should be discussed. Indeed, the previous variants were lethal, and pathways were associated with pathogenicity, overt immune response and so on, so pathways associated with recovery, presented in the reviewed manuscript, are unique to Omicron and the differences are important and should be discussed.

Response: We have added a new section to the Discussion that covers the limitations of our study, and where we also reference several other studies in this mouse model. We also explain that the distinct methodologies make detailed comparisons difficult. We also cover the issue of lethality, which is generally viewed as being due to fulminant brain infections. Transcriptional signatures at early time points are broadly comparable in K18-hACE2 mice for BA.1 versus an original isolate, however, detailed analysis of differences between BA.1 and an original strain isolate study (where methodologies are very similar) are highlighted in Fig. 4b. We have also added a series of new references comparing BA.1 with other isolates in a range of rodent models. 

Comment #2: 2. Analyses of the infected lungs are based on lobes rather than whole lung. Unless the authors demonstrate the homogeneity of infection and of histological changes throughout the lung, taking parts (lobes) of the lungs for different analyses is not reliable.

Response: We have provided a new S1a Fig. to show that despite different assay methods and different lung lobes being used, correlation was very high (r=0.914, p=0.000005) for viral loads measured using RNA-Seq (viral reads cpm) vs. viral titers (log10CCID50). Importantly, very low read counts always corresponds with undetectable viral titers. Variance between mice was also not exceeded by variance between lobes. 

Comment #3: 3. Fig. 1b&c –Viral load data were determined by RNA seq data and are presented as reads per million counts. In b the vertical axis is misleading (1,5,10 than 10 again and then 1E4.) 

Response: We have removed the split axis.

Comment #4: These results are hard to translate to viral load/copy and do not account for genome/antigenome copies and it is recommended to apply real-time RT-PCR to covert the data to genome equivalents. Furthermore, the data in 1c show about 1 log higher copies compared to 1b – whereas the authors indicate that the loads at day 2 post infection are comparable (line 208).

Response: Fig. 1c has now been removed and references to previous data moved to the Discussion as requested (see 4 below). The key general contention, supported by a range of publications (now provided), is that BA.1 lung viral loads are lower and reduce faster than those seen after infection with original strain isolates. Our data is consistent with these previous reports for such comparisons in rodent models, with comparisons supported by viral titer and/or viral read data from RNA-Seq. These two measures provide independent viral load information. Side by side BA.1 vs. original strain qPCR evaluations of lung viral loads over time would provide suitable data, but would not change the aforementioned contention which is now very widely supported by a range of studies and a number of groups. Such evaluations also represents a considerable burden of work and sizable new animal experiments, which we feel are hard to justify in the face of such a body of existing evidence.

Comment #5: 4. The data in graph 1c is from reference #53 and while properly reported as such, should not be part of the results (1c, lines 227-8). These data should be discussed in the discussion.

Response: This graph has been removed and the data discussed in the discussion. 

Comment #6: 5. Fig. 1 d-e – while the authors demonstrate a viral load of about 1E8 CCID50/gr lung (about 2E7/ lung) at day 2, the immunohisthology shows at the same day very few stained areas. Unless image of the whole lung/lobe shows widespread viral antigens throughout the lung to account for the high viral load, the data are inconsistent.

Response: We would argue that IHC is actually quite consistent with viral titers, with one 4 μm section of lung showing robust staining in bronchial epithelial cells. Estimates (likely an underestimate) suggest each infected cell can produce 10-100 infectious SARS-CoV-2 particles in vivo (PNAS 118 (25) e2024815118). We estimate that a mouse lung contains ~2x108 cells (60,000 cells per lung lobe section calculated using QuPath, ~1000 4 μm sections per lung lobe, times 4 lung lobes = ~2x108). Thus to produce a virus titer of 108 CCID50/g (Fig .1), which is thus ~1.6x107 per lung given a mouse lung weighs ~0.16 g, we would expect ~1.6x105 to 1.6x106 infected cells per lung, which is approximately 0.1-1% of all cells infected. This estimate is broadly consistent with our IHC, and also seems to us to be quite consistent with our previous use of this monoclonal antibody in these models; specifically, Yan et al 2022 Virus Evolution 8(2):veac063; Morgan et al 2023 Viruses 2023, 15, 139; Rawle et al 2021 PLoS Pathog 17(7): e1009723; and Carolin et al 2024 doi: https://doi.org/10.1101/2024.05.29.59639. 

Comment #7: 6. Fig. 2b and lines 268-281 – The analyses presented in 2b shows inconsistent yet significant effects following infection with the replicating virus (red) up to day 10. This should be discussed.

Response: We presume that discussion is sought with regard to the apparently inconsistent drop at 5 dpi and/or the rise at 10 dpi. Perhaps pertinent is that data (red) for 2, 5 and 10 dpi (Fig. 2b) are not significantly different (p>0.143). We have thus added the following to the figure legend (For BA.1 infected mice, data for 2, 5 and 10 dpi are not significantly different, while 30 dpi shows a significant reduction from 5 and 10 dpi by t tests, p=0.0.019 and p=0.039, respectively). As this now more clearly illustrates a simple waning of infiltrates with time from 10 dpi, we feel that no particular discussion is warranted.

Comment #8: 7. Fig. 4 – The Z score describes only one aspect of the analysis (e.g. activation/inhibition). -Log P value should be added to demonstrate the robustness of the observed responses.

Response: P values are now provided in an additional figure item for Fig. 4a. All significance values for the bioinformatic treatments are also provided in the Supplementary tables.

Comment #9: 8. Fig. 6b – The data in the three “boxes” seems to be provided in different ways – either p for the two right boxes and -log P value for the left box. It should be corrected such that the values are presented in a similar way.

Response: These represent the statistical outputs provided by the IPA program (Qiagen). The significance calculations are distinct for the canonical pathway annotations, and are thus presented in a slightly different way. Although it would be easy to simply unlog these values, we are reluctant to undertake this change, as it leaves readers with the impression that we are using statistical calculations that are somehow different from those provided by the IPA output. It is also conventional to present IPA statistics in this way (see Bishop et al 2022, 2024).

Comment #10: 9. Fig. 6 c&d – R2 should also be provided.

Response: Coefficients of determination have been added to Fig. 6 c and d as requested.

Comment #11: 10. RNA seq. data should be validated.

Response: Presumably this means by RT qPCR- we have provided such validation for 3 genes for 2 time points (from 30 samples) and present this data into a new S4a Fig. Correlations for log2 fold change for the 3 genes (2 time points) was very high (r = 0.98, p=0.00035, n=6). A method section for RTqPCR has also been added.

Comment #12: 11. Discussion – lines 491-494 – The authors mention that long-COVID features were not identified but with the limited number of mice and the frequency of long-COVID in the population their expectation is unreasonable. The discussion should be corrected.

Response: We sought here simply to state that we found no evidence of any of the manifestations associated with COVID-19 sequelae. To clarify, we have added the following at the end of this Discussion point; “Only 8 mice were analyzed at > 30 dpi, which is arguably too low to identify a potential subset of individuals with post-acute or long-COVID”.

Comment #13: Minor:

1. Detailed description of RNA isolation and preparation of libraries are missing. Placing organs in RNA later is not sufficient.

Response: We have expanded this section to read “RNA isolation, library preparation and RNA-Seq was undertaken as described {Bishop, 2023 #437}{Bishop, 2022 #1}. Briefly, lung tissues were harvested into RNAlater, RNA was extracted using TRIzol (Life Technologies), and RNA concentration and quality measured using TapeStation D1kTapeScreen assay (Agilent). cDNA libraries were generated using Illumina TruSeq Stranded mRNA library prep kit”.

Comment #14: 2. Line 262 – One lobe is not “total lung”.

Response: The word “total” has been removed

---

## [Decision Letter · Decision Letter 1]

28 Aug 2024

PONE-D-24-09241R1Tracking inflammation resolution signatures in lungs after SARS-CoV-2 omicron BA.1 infection of K18-hACE2 micePLOS ONE

Dear Dr. Suhrbier,

Thank you for submitting your manuscript to PLOS ONE. After careful consideration, we feel that it has merit but does not fully meet PLOS ONE’s publication criteria as it currently stands. Therefore, we invite you to submit a revised version of the manuscript that addresses the points raised during the review process.

Please address the minor revision requested by Reviewer #2 regarding Figure 1.

We look forward to receiving your revised manuscript.

Kind regards,

Engin Berber, D.V.M., Ph.D.

Academic Editor

PLOS ONE

Journal Requirements:

Reviewers' comments:

Reviewer's Responses to Questions

**Comments to the Author**

1. If the authors have adequately addressed your comments raised in a previous round of review and you feel that this manuscript is now acceptable for publication, you may indicate that here to bypass the “Comments to the Author” section, enter your conflict of interest statement in the “Confidential to Editor” section, and submit your "Accept" recommendation.

Reviewer #1: All comments have been addressed

Reviewer #2: (No Response)

2. Is the manuscript technically sound, and do the data support the conclusions?

Reviewer #1: Yes

Reviewer #2: Yes

3. Has the statistical analysis been performed appropriately and rigorously? 

Reviewer #1: Yes

Reviewer #2: Yes

4. Have the authors made all data underlying the findings in their manuscript fully available?

Reviewer #1: Yes

Reviewer #2: Yes

5. Is the manuscript presented in an intelligible fashion and written in standard English?

Reviewer #1: Yes

Reviewer #2: Yes

6. Review Comments to the Author

Reviewer #1: The research investigates the dynamics of SARS-CoV-2 omicron BA.1 infection in K18-hACE2 mice, focusing on lung infection and its subsequent clearance. Mice were intranasally inoculated with either live BA.1 virus or UV-inactivated BA.1 virus. Lung tissue was harvested at various days post-infection (dpi) to assess viral titers, RNA levels, and histopathology. Initial results indicated that by 2 dpi, lung viral titers peaked and then significantly declined by 10 dpi, where most mice showed no detectable virus. Viral RNA levels and immunohistochemistry confirmed active infection initially but showed significant reduction in viral load over time. Histological analysis of the lungs showed that infection-induced damage was less severe compared to infections with original strain isolates, with most histopathological features resolving by 66 dpi. The study highlights the concept of “protective inflammation” which is the effectiveness of inflammation in clearing an infection leading to little to no disease, and highlights the fact that most publications focus on the outcomes of severe disease. This is one of the strengths of this manuscript.

The authors have addressed the reviewers concerns leading to an improved manuscript.

Reviewer #2: Dear editor,

I reviewed the revised manuscript By Carolin et al (Review of revised PONE-D-24-09241) and read the authors responses.

While I am satisfied with most of the corrections I would suggest that an additional issue be addressed:

The first revision included the following comment and the response follows:

Comment #4: These results are hard to translate to viral load/copy and do not account for genome/antigenome copies and it is recommended to apply real-time RT-PCR to covert the data to genome equivalents. Furthermore, the data in 1c show about 1 log higher copies compared to 1b – whereas the authors indicate that the loads at day 2 post infection are comparable (line 208).

Response: Fig. 1c has now been removed and references to previous data moved to the Discussion as requested (see 4 below). The key general contention, supported by a range of publications (now provided), is that BA.1 lung viral loads are lower and reduce faster than those seen after infection with original strain isolates. Our data is consistent with these previous reports for such comparisons in rodent models, with comparisons supported by viral titer and/or viral read data from RNA-Seq. These two measures provide independent viral load information. Side by side BA.1 vs. original strain qPCR evaluations of lung viral loads over time would provide suitable data, but would not change the aforementioned contention which is now very widely supported by a range of studies and a number of groups. Such evaluations also represents a considerable burden of work and sizable new animal experiments, which we feel are hard to justify in the face of such a body of existing evidence.

My response (and request) is as follows:

The idea was not to further demonstrate that viral titers decline rapidly in BA.1 infected mice and not to compare to the original strain but rather to use a reliable tool (qRT-PCR) for the data presented now in Fig. 1b . Instead of providing the RNA seq data of viral load analyses (in counts per million – a measure that cannot be translated to genomes/copies) – to analyze the same RNA (no need for more animals or BSL3 facility) by qRT-PCR to provide the data as viral copies/gr thus allowing a better comparison between Fig. 1 a and b.

7. PLOS authors have the option to publish the peer review history of their article (what does this mean?). If published, this will include your full peer review and any attached files.

Reviewer #1: No

Reviewer #2: No

---

## [Author Response · Author response to Decision Letter 1]

2 Oct 2024

Reviewer #2: 

"Dear editor,

I reviewed the revised manuscript By Carolin et al (Review of revised PONE-D-24-09241) and read the authors responses.\\

While I am satisfied with most of the corrections I would suggest that an additional issue be addressed:

The first revision included the following comment and the response follows:

Comment #4: These results are hard to translate to viral load/copy and do not account for genome/antigenome copies and it is recommended to apply real-time RT-PCR to covert the data to genome equivalents. Furthermore, the data in 1c show about 1 log higher copies compared to 1b – whereas the authors indicate that the loads at day 2 post infection are comparable (line 208).

Response: Fig. 1c has now been removed and references to previous data moved to the Discussion as requested (see 4 below). The key general contention, supported by a range of publications (now provided), is that BA.1 lung viral loads are lower and reduce faster than those seen after infection with original strain isolates. Our data is consistent with these previous reports for such comparisons in rodent models, with comparisons supported by viral titer and/or viral read data from RNA-Seq. These two measures provide independent viral load information. Side by side BA.1 vs. original strain qPCR evaluations of lung viral loads over time would provide suitable data, but would not change the aforementioned contention which is now very widely supported by a range of studies and a number of groups. Such evaluations also represents a considerable burden of work and sizable new animal experiments, which we feel are hard to justify in the face of such a body of existing evidence.

My response (and request) is as follows:

The idea was not to further demonstrate that viral titers decline rapidly in BA.1 infected mice and not to compare to the original strain but rather to use a reliable tool (qRT-PCR) for the data presented now in Fig. 1b . Instead of providing the RNA seq data of viral load analyses (in counts per million – a measure that cannot be translated to genomes/copies) – to analyze the same RNA (no need for more animals or BSL3 facility) by qRT-PCR to provide the data as viral copies/gr thus allowing a better comparison between Fig. 1 a and b."

Response:

We have provided a side-by-side comparison between BA.1 and an original strain isolate using a RTqPCR based method to generate genome copy number data, as requested. This data is presented in a new Fig. 1c, with new sections for methods and results.

---

## [Editor Report · Decision Letter 2]

18 Oct 2024

Tracking inflammation resolution signatures in lungs after SARS-CoV-2 omicron BA.1 infection of K18-hACE2 mice

PONE-D-24-09241R2

Dear Dr. Suhrbier,

We’re pleased to inform you that your manuscript has been judged scientifically suitable for publication and will be formally accepted for publication once it meets all outstanding technical requirements.

Kind regards,

Engin Berber, D.V.M., Ph.D.

Academic Editor

PLOS ONE
---

## [Editor Report · Acceptance letter]

23 Oct 2024

PONE-D-24-09241R2 

PLOS ONE

Dear Dr. Suhrbier, 

I'm pleased to inform you that your manuscript has been deemed suitable for publication in PLOS ONE. Congratulations! Your manuscript is now being handed over to our production team.

Kind regards, 

on behalf of

Dr. Engin Berber 

Academic Editor

PLOS ONE